# Large Gender Gap in Oral Hygiene Behavior and Its Impact on Gingival Health in Late Adolescence

**DOI:** 10.3390/ijerph17124394

**Published:** 2020-06-18

**Authors:** Masanobu Abe, Akihisa Mitani, Kazuto Hoshi, Shintaro Yanagimoto

**Affiliations:** 1Division for Health Service Promotion, The University of Tokyo, Tokyo 113-0033, Japan; mitania-int@h.u-tokyo.ac.jp (A.M.); yanagimoto@hc.u-tokyo.ac.jp (S.Y.); 2Department of Oral & Maxillofacial Surgery, The University of Tokyo Hospital, Tokyo 113-8655, Japan; hoshi-ora@h.u-tokyo.ac.jp

**Keywords:** oral hygiene behavior, sex, gender, periodontal disease, adolescence, adolescent

## Abstract

*Background*: Epidemiologic studies provide broad-based evidence that males are at greater risk of severe periodontal diseases than females. Our recent findings further revealed that male gender was an independent risk factor for gingival bleeding in late adolescents in Japan. Gingival health status has been reported to be affected by oral hygiene behavior. However, gender difference in this regard has not yet been clarified. *Methods*: We conducted a retrospective review of mandatory medical questionnaires administered as part of a legally required freshman medical checkup between April 2017 and 2019 at the University of Tokyo. *Results*: Among a total of 9376 sets of responses, chosen subjects were 9098 students aged 17–19. For frequency of daily brushing, males brushed less frequently than females (*p* < 0.001): 1 time or less (male: 22.9%, female: 11.2%), twice (65.0%, 69.2%), 3 times or more (12.1%, 19.6%). For the duration of brushing each time, males brushed for a shorter period of time than females (*p* = 0.005): 1 min or less (male: 17.2%, female: 14.1%), 2–3 min (46.9%, 49.2%), 4 min or more (35.9%, 36.7%). (1) Male gender, (2) lower frequency of daily brushing and (3) shorter duration of brushing each time, were significantly associated with the presence of gingival bleeding (*p* < 0.001 for all). Multivariate regression analysis showed that: (1) male gender (odds ratio 1.29, 95% confidence interval 1.15–1.44); (2) frequency of daily brushing: 1 time or less (2.36, 2.02–2.76), twice (1.45, 1.27–1.67); and (3) brushing duration each time: 1 min or less (1.57, 1.39–1.78), 2–3 min (1.26, 1.14–1.39), were independent risk factors for gingival bleeding (*p* < 0.001 for all). *Conclusions*: Males showed poorer oral hygiene behavior than females in late adolescents in Japan. Male gender was an independent risk factor for gingival bleeding, as well as poor oral hygiene behavior.

## 1. Introduction

Periodontal diseases (gingivitis and periodontitis) are highly prevalent, affecting up to 90% of the worldwide population [1]. They have been associated with various systemic diseases such as heart diseases, diabetes, respiratory diseases, rheumatism, metabolic syndrome, and so on [2,3,4,5,6,7,8,9,10,11]. Even in adolescents, the presence of gingival bleeding, one of the major symptoms of periodontal diseases, is suggested to be associated with several systemic diseases [12]. Therefore, the prevention and treatment of periodontal diseases have become increasingly important. 

Large, cross-sectional, epidemiologic studies provide broad-based evidence of a sexual dimorphism in destructive periodontal diseases, reflecting a greater prevalence of periodontitis in males than females [13,14]. Our recent research adds a new insight to these findings; that is, male gender is an independent risk factor for gingival bleeding in late adolescents [12]. It has been reported that gingival health status is affected by oral hygiene behavior [15,16,17,18,19,20]. However, gender differences in oral hygiene behavior have not yet been clarified. The objective of this study was to investigate differences in oral hygiene behavior and its impact on gingival health in late adolescence by gender.

## 2. Methods

### 2.1. Study Design and Population

We performed a retrospective review of mandatory medical questionnaires administered as a part of a legally required freshman medical checkup between April 2017 and April 2019 at the University of Tokyo. Among a total of 9376 responses (7563 males, 1813 females, aged 17–59 years, mean age 18.4 years), all of the responses from students aged less than 20 years (9098 sets, 7316 males, 1782 females aged 17–19 years, mean age 18.3 years) were subjected to analyses.

### 2.2. Questionnaire

A self-administered, closed-ended questionnaire was distributed to all students (Appendix A). To assess frequency of toothbrushing, the questionnaire asked: “How many times do you brush a day? Please choose one of the following options; 0 time, 1 time, 2 times, 3 times, 4 times or more“. Another question addressed the duration of toothbrushing: “How long do you brush each time? Please choose one from the following options; less than 1 min, 1 min, 2–3 min, 4–5 min, 6 min or more“. Gingival health status was assessed with the yes/no question: “Are you aware of gum bleeding when you brush?” [12]. After tallying each student’s responses to the questionnaire, we investigated differences in the status of oral hygiene behavior between male and female adolescents. 

### 2.3. Statistical Analyses

Statistical analysis was performed using a χ^2^ test and multivariate binomial logistic regression models. A value of *p* < 0.05 (two-sided) was accepted as significant. All the analyses were conducted using the statistical software program SAS ver. 9.4 (SAS Institute Inc., Cary, NC, USA).

### 2.4. Ethical Approval

This retrospective study was approved by the research ethics committee of the University of Tokyo in 2018, approval no. 18-197 (currently, revised as no. 19-324), “Retrospective analyses of medical and health record information retained by the division for health service promotion, the University of Tokyo.” We abided by all relevant laws, regulations and the university rules for privacy.

## 3. Results

### 3.1. Gender Difference for the Frequency of Daily Brushing in Late Adolescents

The data of 9098 students aged 17–19 were analyzed. Of these, data from 9072 students were valid for analysis of daily frequency of brushing. The respondents were categorized into three groups by their brushing frequency: “1 time or less”, “twice”, “3 times or more”. In both males and females, brushing twice a day was the most common frequency. There was a greater number of males than females in the”1 time or less” group. The was a greater number of females than males in the other two groups (Figure 1, Table 1). The χ^2^ test revealed that males brush less frequently than females (*p* < 0.001). 

### 3.2. Gender Difference for the Duration of Brushing Each Time in Late Adolescents 

Of data from 9098 students aged 17–19, those from 9070 students were valid for the analysis of duration of brushing each time. The respondents were categorized into three groups by the duration of brushing each time: “1 min or less”, “2–3 min”, and “4 min or more”. Both in males and females, “2–3 min” was the most common brushing duration. There was a greater number of males than females in the group of “1 min or less”. There was a greater number of females than males in the other two groups (Figure 2, Table 2). The χ^2^ test revealed that males brush for shorter periods of time than females (*p* = 0.005).

### 3.3. The Association of Gingival Bleeding with Gender and Oral Hygiene Behavior

The associations of gingival bleeding with gender and oral hygiene behavior (frequency of daily brushing and brushing duration) were analyzed by χ^2^ tests. The results revealed that male gender, lower frequency of daily brushing and shorter brushing duration were significantly associated with the presence of gingival bleeding (*p* < 0.001 for all, Table 3). A multivariate analysis was conducted for the complete series of respondents using a binomial logistic regression model with gingival bleeding as the objective variable (gingival bleeding as an event), and using male gender, frequency of daily brushing (≤1time, 2 times), and brushing duration (≤1 min, 2–3 min) as explanatory variables. The results showed significant rates for the following factors: (1) male gender (odds ratio (OR) 1.29, 95% confidence interval (CI): 1.15–1.44, *p* < 0.001), (2) frequency of daily brushing: ≤1 time (OR 2.36, 95% CI: 2.02–2.76, *p* < 0.001), 2 times (OR 1.45, 95% CI: 1.27–1.67, *p* < 0.001) and (3) brushing duration: ≤1 min (OR 1.57, 95% CI: 1.39–1.78, *p* < 0.001), 2–3 min (OR 1.26, 95% CI: 1.14–1.39, *p* < 0.001). Male gender, frequency of daily brushing (≤1time, 2 times), and duration of brushing each time (≤1 min, 2–3 min), were independent risk factors for gingival bleeding (Table 4).

## 4. Discussion

Periodontal diseases, gingivitis and periodontitis are the most common oral disorders affecting the supporting structures of teeth and are partially responsible for tooth loss. They are also associated with various systemic diseases [2,3,4,5,6,7,8,9,10,11,12,21,22]. Thus, prevention and treatment of periodontal diseases have recently become a public health focus [23,24,25].

Lower frequency of daily brushing and shorter brushing duration have been closely associated with gingival bleeding. In multivariate analysis, they were found to be independent risk factors for gingival bleeding. There have been relatively fewer studies evaluating the association between oral hygiene behaviors and periodontal diseases. However, a recent meta-analysis indicated that infrequent tooth brushing was associated with severe forms of periodontal disease [19]. In another multivariate analysis, Lertpimonchai et al. revealed that poor oral hygiene behavior increased the risk of periodontitis [20]. In this study, we increased the evidence level of the association between oral hygiene behavior and periodontal disease. Periodontal diseases occur by a complex interplay of various etiological factors. Particularly, a history of medication can cause gingival bleeding. Thus, we would like to address this issue in our future study.

In the current study, males displayed poorer oral hygiene behavior than females. We found similar results that revealed gender difference with regard to daily toothbrushing frequency in our previous study involving university students [18]. Outside Japan, Olczak-Kowalczyk et al. investigated oral hygiene status in Polish adolescents and found that prevalence of gingival bleeding was higher in males [16]. Almas et al. revealed that female university students had better oral hygiene practices compared to male students in Saudi Arabia [26]. Even in adults, significant gender difference for the frequency of daily brushing has been reported [27]. Thus, the significant gender difference in oral hygiene behavior observed in this study might point to a common problem worldwide regardless of age. In a future study, we would like to address other parameters such as education level, economic status, status of usage of other oral hygiene devices such as dental floss, electric toothbrush, interdental brush, mouthwash and tongue cleaner in late adolescents.

Large epidemiologic studies provide evidence that severe periodontal disease has a documented higher prevalence in males compared to females [13,14]. In this study, male gender was an independent risk factor for gingival bleeding as well as poorer oral hygiene behavior in late adolescents. The result suggested that the gender gap of morbidity for severe periodontal diseases, which are generally observed in adults, had already begun in late adolescence. Sexual dimorphisms sometimes result from genetic and epigenetic differences in gene regulation [28]. Differential gene expression in sex steroid-responsive genes were reported to contribute to a sexual dimorphism in susceptibility to severe periodontal disease [14].

Systemic health condition is affected by oral hygiene level. For example, improved oral hygiene behavior is associated with decreased risk of atrial fibrillation and heart failure [29]. Periodontal therapy provides good systemic effects in patients with type 2 diabetes [22]. A meta-analysis showed that the risk of periodontitis could be reduced by regular toothbrushing [20]. Thus, increased awareness of oral health and hygiene behavior in young people, particularly males, would decrease not only the future severity of periodontal diseases but also systemic diseases that are affected by gingival health status in the elderly.

In summary, it was found that males showed poorer oral hygiene behavior than females in late adolescents in Japan and, furthermore, male gender was an independent risk factor for gingival bleeding as well as poor oral hygiene behavior. However, there was a disparity in the recruited number of male and female subjects in the present study. Thus, additional studies are needed to validate these findings by equal recruitment of male and female subjects.

## 5. Conclusions

Males showed poorer oral hygiene behavior than females in late adolescents in Japan. Male gender was an independent risk factor for gingival bleeding as well as poor oral hygiene behavior. Further research is needed to validate these findings.

## Figures and Tables

**Figure 1 ijerph-17-04394-f001:**
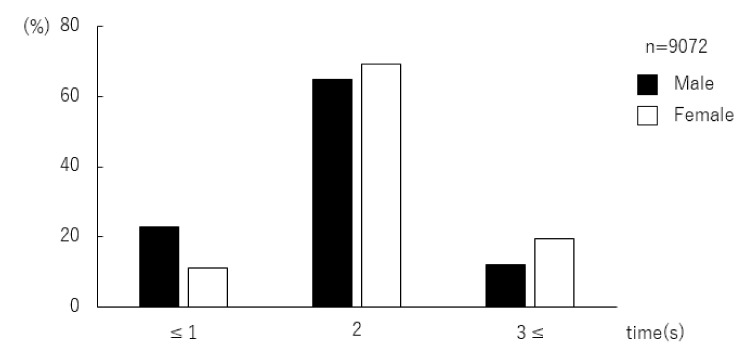
Distribution of adolescents by frequency of daily brushing. In both males and females, brushing “twice” a day was the most common frequency. There was a greater number of males than females in the group “1 time or less” a day. There was a greater number of females than males in the other two groups, “twice” and “3 times or more” a day.

**Figure 2 ijerph-17-04394-f002:**
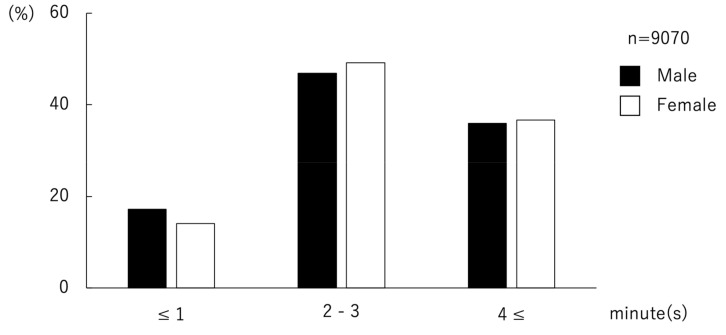
Distribution of adolescents for duration of toothbrushing. In both males and females, “2 or 3 min” was the most common duration of brushing each time. There was a greater number of males than females in the group “1 min or less”. The was a greater number of females than males in the other two groups, “2 or 3 min” and “4 min or more”.

**Table 1 ijerph-17-04394-t001:** Gender difference in daily toothbrushing frequency (χ^2^ test).

Frequency of Daily Brushing	All (*n* = 9072)	Male (*n* = 7294)	Female (*n* = 1778)	*p*
Time (s)	*n* (%)	*n* (%)	*n* (%)	
≤1	1868 (20.6)	1669 (22.9)	199 (11.2)	<0.001 *
2	5971 (65.8)	4741 (65.0)	1230 (69.2)	
3≤	1233 (13.6)	884 (12.1)	349 (19.6)	

*: <0.05.

**Table 2 ijerph-17-04394-t002:** Gender difference in toothbrushing duration (χ^2^ test).

Duration of Brushing	All (*n* = 9070)	Male (*n* = 7293)	Female (*n* = 1777)	*p*
min	*n* (%)	*n* (%)	*n* (%)	
≤1	1508 (16.6)	1258 (17.2)	250 (14.1)	0.005 *
2–3	4291 (47.3)	3417 (46.9)	874 (49.2)	
4≤	3271 (36.1)	2618 (35.9)	653 (36.7)	

*: <0.05.

**Table 3 ijerph-17-04394-t003:** The association of gingival bleeding with gender and oral hygiene behavior (χ^2^ test).

	*n*	Gingival Bleeding		*p*
		Presence	Absence	
		*n* (%)	*n* (%)	
All	9098	3321 (36.5)	5777 (63.5)	
*Gender*				
Male	7316	2780 (38.0)	4536 (62.0)	<0.001 *
Female	1782	541 (30.4)	1241 (69.6)	
*Frequency of daily brushing* (times)				
≤1	1868	881 (47.2)	987 (52.8)	<0.001 *
2	5971	2108 (35.3)	3863 (64.7)	
3≤	1233	331 (26.8)	902 (73.2)	
*Duration of brushing* (min)				
≤1	1508	653 (43.3)	855 (56.7)	<0.001 *
2–3	4291	1593 (37.1)	2698 (62.9)	
4≤	3271	1073 (32.8)	2198 (67.2)	

*: <0.05.

**Table 4 ijerph-17-04394-t004:** Multivariate regression analysis for the association of gingival bleeding with gender and oral hygiene behavior.

	Odds Ratio (95% CI)	*p*
*Gender*		
Male	1.29 (1.15–1.44)	<0.001 *
Female	1	–
*Frequency of daily brushing* (times)		
≤1	2.36 (2.02–2.76)	<0.001 *
2	1.45 (1.27–1.67)	<0.001 *
3≤	1	–
*Duration of brushing* (min)		
≤1	1.57 (1.39–1.78)	<0.001 *
2–3	1.26 (1.14–1.39)	<0.001 *
4≤	1	–

CI: confidence interval, *: <0.05.

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
