# Peer review of "Large Gender Gap in Oral Hygiene Behavior and Its Impact on Gingival Health in Late Adolescence"

_ijerph, 2020, doi:10.3390/ijerph17124394_

Round 1
Reviewer 1 Report
The manuscript submitted to IJERPH entitled “Large Gender Difference in Oral Hygiene Behavior and Its Impact on Gingival Health in Japanese Adolescents” is an original article which aim to investigate possible relation between gender and gingival health in adolescents.
On my opinion the article is interesting, well written, with good English. The content of the manuscript is very interesting. The authors showed relationship between gender and gingival health in a student population of Japan.
However, I highlighted some critical issues.
Title. Please modify the title: In my opinion it would be better to reduce the number of words and be clearer about what the authors investigated
Abstract. It may be helpful to reduce the number of words to attract the reader's attention.
Introduction. Are there studies in the literature concerning diseases in the student population? Better specify the objectives and methods of the study.
Methods. Is it possible to insert a copy of the questionnaire as supplementary material? Specify the year of approval of the ethics committee. Is it a university or hospital ethics committee?
Results. Have students been diagnosed with periodontal disease?
Discussion. Are there other similar studies that have shown similar results in the adult population? Did the authors find limitations in their study by comparing it with other in the literature?
Conclusion. Please improve. The interpretation of the results is not clear. No study limits are mentioned.
I don't feel qualified to judge about the English language and style.
There is no conflict of interest between me and any of the authors.
Author Response
We appreciate this reviewer's precise and constructive comments.
Comments:
- Please modify the title: In my opinion it would be better to reduce the number of words and be clearer about what the authors investigated
Owing to the comment, the title of the manuscript was modified.
- It may be helpful to reduce the number of words to attract the reader's attention.
Reflecting this comment, we tried to reduce the number of words in Abstract section. However, a reviewer recommended to put detail data, so we could not reduce the number of words. We are sorry about that.
- Are there studies in the literature concerning diseases in the student population? Better specify the objectives and methods of the study.
Owing to the comment, we discussed about it in Introduction section and specified the objectives of this study.
- Is it possible to insert a copy of the questionnaire as supplementary material? Specify the year of approval of the ethics committee. Is it a university or hospital ethics committee?
We prepared an additional file for the questionnaire (it was originally written in Japanese). This study was approved by the research ethics committee of University. The year of approval of the ethics committee was specified (Method section).
- Have students been diagnosed with periodontal disease?
As the reviewer pointed out, the students have not been diagnosed with periodontal diseases. Thus, we carefully use the word "gingival bleeding (a major symptom of periodontal diseases)" instead of periodontal diseases through manuscript.
- Are there other similar studies that have shown similar results in the adult population? Did the authors find limitations in their study by comparing it with other in the literature?
We found a study which revealed significant gender difference for the frequency of daily brushing in adults (Gorska R et al., Dent Med Probl, 2018). However, the impact of the frequency on gingival health by gender has not been addressed in the study. The manuscript was cited and discussed in the Discussion section.
- Please improve. The interpretation of the results is not clear. No study limits are mentioned.
We improved the Conclusion to make the interpretation of the results clear. A paragraph of study limitations was included in the Discussion section.
Reviewer 2 Report
Dear Authors,
This retrospective study conducted on male and female Japanese adolescents reveals the differences in oral hygiene behavioral patterns and their impact on gingival health among the male and female subjects. The study results indicate that males adopted poorer oral hygiene practices than females in terms of frequency of daily brushing and duration of brushing time. Correlating similar findings from previous studies and the results of the current study, the authors speculate that the male gender is an independent risk factor for gingivitis in young adolescents.
The results of the study instigate future research to explore further the proposed hypotheses.
However, the authors should attend to the following concerns to enhance the suitability of the paper for publication.
- Under Abstract - Line 8: 'subjects were..' - can be changed to 'chosen subjects were..'
- Under section 3.1- Line 7: '..males brush their teeth..' - can change to '..males brushed..' (it is good to stick on to the same tense throughout the manuscript)
- Authors in the current study have substantiated that the frequency of brushing and duration of brushing are two main parameters taken into consideration to assess oral hygiene behavior. However, in future studies do the authors consider including any other significant parameters as well to assess the oral hygiene behavior of an individual to strongly establish the proposed hypotheses? It's crucial to highlight this under 'Discussion'.
- Out of the 9072 study subjects recruited for the study, there is a tremendous disparity between male (n=7294) and female (n=1778) subjects. In order to come up with conclusive results such as 'male gender is an independent risk factor for gingival bleeding and periodontal disease in adolescents', an attempt to conduct a study on a population that involves almost equal recruitment of male and female subjects will be more meaningful. As this is a study that talks about the large gender difference for a particular symptom of periodontal disease, eliminating this bias would add more significance to the conclusions drawn.
- Under discussion - Line 6: 'In multivariate...independent risk factors for it' - please consider using the right term instead of using 'it' (sentence seems incomplete)
- Under discussion - Line 7: 'relatively few..' - change to 'relatively fewer..'
- Under discussion - Line 12: "We recommend '3 or more times' for the frequency of brushing and '4 mins and more' brushing duration to prevent the onset of periodontal disease". This particular statement needs more validation and supporting data as of why do the authors exactly recommend '3 or more times' brushing and why not '2 or more times' etc. The authors may attempt to provide evidence for this specific recommendation from previous similar studies in order to support their statement.
- Page 6 - Line 3 & 4: can consider rephrasing the two sentences to have better continuity. For instance, can consider modifying as - 'In the current study males displayed/presented poorer oral hygiene behavior than females. Interestingly, we found similar results for the gender difference of daily toothbrushing frequency in our previous study that involved students of Japenese University'.
- Page 6 - Line 8: "The gender difference in oral hygiene behavior observed here might point to a common problem worldwide, to a greater or lesser extent" - authors are advised to be more specific when expressing such statements and replace the words such as 'here' with the right terms and may also consider having better continuity with the previous statements.
- Page 6 - Line 12: ".. an independent risk factor of.." - change to "..an independent risk factor for.."
- The results are well expressed in terms of multivariate regression analysis and p-Value demonstrating statistical significance for the two parameters studied. However, additionally, the expression of the results/data in terms of percentage differences will make the outcomes of the study more absorbable for the readers. In fact, under the abstract, interpreting the results by showing percentage differences can give the readers a quick understanding of the study results and also intrigue the readers to read through the paper further.
- Gingivitis and periodontitis occur owing to a complex interplay of various etiological factors. In order to establish poor oral hygiene behavior as an independent risk factor for gingival bleeding, authors need to consider mentioning the exclusion criteria as well. Are all the subjects systemically healthy, any history of medication (certain drugs can cause gingival bleeding), history of parafunctional habits? Further, a complete assessment of gingival bleeding would involve a physical examination besides just a subjective response with only one question of "are you aware of bleeding gums when brushing your teeth?"
Author Response
We appreciate this reviewer's precise and constructive comments.
Comments:
- Under Abstract - Line 8: 'subjects were..' - can be changed to 'chosen subjects were..'
Owing to the comment, it was modified.
- Under section 3.1- Line 7: '..males brush their teeth..' - can change to '..males brushed..' (it is good to stick on to the same tense throughout the manuscript)
Reflecting this comment, we used '..males brushed..' instead of '..males brush their teeth..' through the manuscript.
- Authors in the current study have substantiated that the frequency of brushing and duration of brushing are two main parameters taken into consideration to assess oral hygiene behavior. However, in future studies do the authors consider including any other significant parameters as well to assess the oral hygiene behavior of an individual to strongly establish the proposed hypotheses? It's crucial to highlight this under 'Discussion'.
We appreciate this reviewer's constructive comments. In future study, we would like to address other parameters such as education level, economic status, status of usage the other oral hygiene devices (dental floss, electric toothbrush, interdental brush, mouthwash, tongue cleaner, etc.). This issue was described in Discussion section.
- Out of the 9072 study subjects recruited for the study, there is a tremendous disparity between male (n=7294) and female (n=1778) subjects. In order to come up with conclusive results such as 'male gender is an independent risk factor for gingival bleeding and periodontal disease in adolescents', an attempt to conduct a study on a population that involves almost equal recruitment of male and female subjects will be more meaningful. As this is a study that talks about the large gender difference for a particular symptom of periodontal disease, eliminating this bias would add more significance to the conclusions drawn.
We appreciate this critical comment. As the reviewer pointed out, almost equal recruitment of male and female subjects is considered to be important. Thus, this was discussed in the Discussion section. We will eliminate this bias in the next validation study.
- Under discussion - Line 6: 'In multivariate...independent risk factors for it' - please consider using the right term instead of using 'it' (sentence seems incomplete)
Owing to the comment, we used “gingival bleeding” instead of “it”.
- Under discussion - Line 7: 'relatively few..' - change to 'relatively fewer..'
Owing to the comment, we changed “relatively few” to “relatively fewer”.
- Under discussion - Line 12: "We recommend '3 or more times' for the frequency of brushing and '4 mins and more' brushing duration to prevent the onset of periodontal disease". This particular statement needs more validation and supporting data as of why do the authors exactly recommend '3 or more times' brushing and why not '2 or more times' etc. The authors may attempt to provide evidence for this specific recommendation from previous similar studies in order to support their statement.
As the reviewer pointed out, the statement needs more validation. Thus, we removed the description from Discussion section.
- Page 6 - Line 3 & 4: can consider rephrasing the two sentences to have better continuity. For instance, can consider modifying as - 'In the current study males displayed/presented poorer oral hygiene behavior than females. Interestingly, we found similar results for the gender difference of daily toothbrushing frequency in our previous study that involved students of Japenese University'.
Reflecting this comment, we modified the sentence.
- Page 6 - Line 8: "The gender difference in oral hygiene behavior observed here might point to a common problem worldwide, to a greater or lesser extent" - authors are advised to be more specific when expressing such statements and replace the words such as 'here' with the right terms and may also consider having better continuity with the previous statements.
Owing to the comment, we modified the sentence as follows: "Thus, the significant gender difference in oral hygiene behavior observed in this study might point to a common problem worldwide regardless of age."
- Page 6 - Line 12: ".. an independent risk factor of.." - change to "..an independent risk factor for.."
Thank you for this comment. We changed "of" to "for" through the manuscript.
- The results are well expressed in terms of multivariate regression analysis and p-Value demonstrating statistical significance for the two parameters studied. However, additionally, the expression of the results/data in terms of percentage differences will make the outcomes of the study more absorbable for the readers. In fact, under the abstract, interpreting the results by showing percentage differences can give the readers a quick understanding of the study results and also intrigue the readers to read through the paper further.
Thank you so much for this suggestion. Percentage differences were shown in Abstract section.
- Gingivitis and periodontitis occur owing to a complex interplay of various etiological factors. In order to establish poor oral hygiene behavior as an independent risk factor for gingival bleeding, authors need to consider mentioning the exclusion criteria as well. Are all the subjects systemically healthy, any history of medication (certain drugs can cause gingival bleeding), history of parafunctional habits?
All of the responses from students aged less than 20 years were subjected to analyses. As the reviewer pointed out, periodontal diseases occur owing to a complex interplay of various etiological factors. Particularly, the history of medication can cause gingival bleeding. Although it has not been asked in the questionnaire, we would like to address this issue in future study. This was described in Discussion section.
- Further, a complete assessment of gingival bleeding would involve a physical examination besides just a subjective response with only one question of "are you aware of bleeding gums when brushing your teeth?"
Students have not been diagnosed with periodontal diseases. Thus, we carefully use the word "gingival bleeding (a major symptom of periodontal diseases)" instead of periodontal diseases through manuscript.
Reviewer 3 Report
Introduction, P2, L5; the reference is not in compliance with the guidelines and should be in number:...(Abe et al)...
It would be better to give more information on stratification according to gender and possible parameters that may be involved in order to explain/show the need for "clarification" of relations between "oral hygiene behavior" and "gender differences".
it is better to use an indirect for of expression like;
"...The objective of this study was to investigate..." instead of "...In this study, we investigated the differences in...".
There should be a definition of patient inclusion and exclusion criteria.
I think after these issues will be addressed, the article will be significantly improved and suitable for publication.
Author Response
We appreciate this reviewer's precise and constructive comments.
Comments:
- Introduction, P2, L5; the reference is not in compliance with the guidelines and should be in number:...(Abe et al)...
Owing to the comment, it was corrected.
- It would be better to give more information on stratification according to gender and possible parameters that may be involved in order to explain/show the need for "clarification" of relations between "oral hygiene behavior" and "gender differences".
As the reviewer pointed out, we should have put more information that might be involved in oral hygiene behavior to evaluate the effect of gender difference on oral hygiene behavior. Study limitation was added in the Discussion section. In our future study, this issue has to be addressed.
- It is better to use an indirect for of expression like;"...The objective of this study was to investigate..." instead of "...In this study, we investigated the differences in...".
Owing to the comment, we modified the sentence.
- There should be a definition of patient inclusion and exclusion criteria.
Reflecting this comment, Methods section was modified as follows; all of the responses from students aged less than 20 years were subjected to analyses.
Round 2
Reviewer 1 Report
The authors edited the manuscript following the reviewers' suggestions. In my opinion it could now be suitable for publication.
Author Response
Comment:
- In my opinion it could now be suitable for publication.
We thank very much for this reviewer's comment.
Reviewer 2 Report
Dear Authors,
Great work! Kindly consider the following minor corrections:
Page 6, Line 4 & 5: For better clarity and continuity, please consider modifying the sentence as follows - "Interestingly, we found similar results that revealed gender difference with regard to daily toothbrushing frequency in our previous study that involved university students".
Page 6, Line 14: change 'status of usage the' to 'status of usage of other..'
Page 6, Line 34 & 35: Please modify 'However, there is a disparity of numbers of between male and female subjects' to "However, there is a disparity in the recruited number of male and female subjects in the present study".
Under conclusion, Line 3: can consider changing 'verify' to 'validate'
Author Response
We appreciate the reviewer's precise comments.
Comments:
Great work! Kindly consider the following minor corrections:
- Page 6, Line 4 & 5: For better clarity and continuity, please consider modifying the sentence as follows - "Interestingly,we found similar results that revealed gender difference with regard to daily toothbrushing frequency in our previous study that involved university students".
Owing to the comment, the sentence was modified.
- Page 6, Line 14: change 'status of usage the' to 'status of usage of other..'
Reflecting this comment, it was modified.
- Page 6, Line 34 & 35: Please modify 'However, there is a disparity of numbers of between male and female subjects' to "However, there is a disparity in the recruited number of male and female subjects in the present study".
Reflecting this comment, the sentence was modified.
- Under conclusion, Line 3: can consider changing 'verify' to 'validate'
Owing to the comment, the word was changed.